# Physicochemical Properties of Plasma-Activated Water and Its Control Effects on the Quality of Strawberries

**DOI:** 10.3390/molecules28062677

**Published:** 2023-03-16

**Authors:** Xiao Yang, Can Zhang, Qunfang Li, Jun-Hu Cheng

**Affiliations:** 1School of Food Science and Engineering, South China University of Technology, Guangzhou 510641, China; 2Academy of Contemporary Food Engineering, South China University of Technology, Guangzhou Higher Education Mega Centre, Guangzhou 510006, China; 3Shanwei Cathay Group, Shanwei 516601, China

**Keywords:** plasma-activated water, strawberry, physicochemical properties, multivariate data analysis

## Abstract

In this study, the effects of plasma-activated water (PAW), generated by dielectric barrier discharge cold plasma at the gas–liquid interface, on the quality of fresh strawberries during storage were investigated. The results showed that, with the prolongation of plasma treatment time, the pH of PAW declined dramatically and the electrical conductivity increased significantly. The active components, including NO_2_^−^, NO_3_^−^, H_2_O_2_, and O_2_^−^, accumulated gradually in PAW, whereas the concentration of O_2_^−^ decreased gradually with the treatment time after 2 min. No significant changes were found in pH, firmness, color, total soluble solids, malondialdehyde, vitamin C, or antioxidant activity in the PAW-treated strawberries (*p* > 0.05). Furthermore, the PAW treatment delayed the quality deterioration of strawberries and extended their shelf life. Principal component analysis and hierarchical cluster analysis showed that the PAW 2 treatment group demonstrated the best prolonged freshness effect, with the highest firmness, total soluble solids, vitamin C, and DPPH radical scavenging activity, and the lowest malondialdehyde and ∆E* values, after 4 days of storage. It was concluded that PAW showed great potential for maintaining the quality of fresh fruits and extending their shelf life.

## 1. Introduction

Over the last few decades, physical techniques have shown great potential for preserving strawberries. These include refrigeration [1,2], irradiation [3], and preheating treatments [4]. However, these technologies have negative influences on the internal and external quality parameters of strawberries, causing serious nutritional loss and even loss in commercial value. Cold plasma technology, as a new and efficient nonthermal processing technology, has gained attention in the field of food preservation. Cold plasma technology makes use of charged, highly reactive gas molecules and substances at ambient temperatures to rapidly act on contaminated microorganisms on food surfaces [5]. It is particularly beneficial for the treatment of heat-sensitive products and the sterilization of internally-packaged foods. Due to the high reactivity of active particles in plasma, they can interact with nearly all cellular components and play a significant role in food quality maintenance [6].

Cold plasma generators suitable for use in food storage and safety can generally be subdivided into atmospheric pressure plasma jet (APPJ) generators and dielectric barrier discharge (DBD) generators. APPJs have their own structural characteristics which can lead to different sterilization effects different distances from the nozzle [7]. As for DBD cold plasma, the entire agricultural product is placed within the plasma discharge region, leading to coverage of all external surfaces. The irregular shape and relative dielectric constant of the product itself can influence the uniform electric field in the reactor where it is placed, resulting in an uneven electric field intensity on the surface of the product [8]. Therefore, the direct action of cold plasma on the surface of food materials usually causes physical damage and limits the application of plasma technology to fresh fruits and vegetables. In recent years, functionalized water, such as plasma-activated water (PAW), as an intermediate medium, was developed to eliminate the limitations of direct cold plasma treatment [9]. Ordinary sterile or distilled water treated by a plasma device is known as PAW. PAW produces numerous short-lived actives, such as ∙OH, ^1^O_2_, and O_2_^−^, as well as long-lived actives, such as H_2_O_2_, O_3_, and NO_3_^−^ [10]. This gives PAW the advantage of broad-spectrum efficient sterilization [11,12]. The inactivation of PAW is the result of a combination of high oxidation type and low pH [13,14]. The homogeneity of the solution and the good fluidity of the liquid can compensate for the inhomogeneity of direct treatment by cold plasma. In addition, these active substances are unstable and can break down quickly. Therefore, no hazardous compounds remain on the surface of agricultural products treated with PAW [15]. As such, PAW has been recognized as having great potential applications in food quality and safety maintenance.

PAW has been confirmed to be effective in microbial inactivation [16,17], seed germination and growth [18,19], meat curing [20], pesticide residue degradation [21,22], and food preservation. PAW has been successfully used to extend the shelf life of fruits, including pears [23], kumquats [24], apples [25], and kiwis [26]. Relatively little information is available, however, on the physicochemical properties of PAW generated by DBD plasma devices and its applications in strawberry preservation.

Therefore, in this study, an atmospheric pressure DBD cold plasma system was used to prepare PAW with a large number of active substances at the gas–liquid interface. The variations of reactive oxygen species (ROS) and reactive nitrogen species (RNS) in PAW, with different treatment times, were investigated. The effects of PAW treatment on the quality of strawberries—in terms of pH, firmness, color, total soluble solids (TSS) content, malondialdehyde (MDA) content, vitamin C (VC) content, and antioxidant activity—were also investigated and reported herein. The study aimed to determine the efficacy of PAW technology for strawberry preservation.

## 2. Results and Discussion

### 2.1. pH and Electrical Conductivity Changes of PAW

As shown in Figure 1a, the pH of PAW decreased as the treatment time increased. Compared with PAW 0, the pH of PAW 1 was significantly decreased, which was caused by the ionization of water producing H^+^ and acidic substances, such as nitrous acid and nitric acid [27]. After 1 min of plasma treatment, the pH of the PAWs tended to decrease more gently, with minimal variation between PAW 5 and PAW 1. The PAW solutions were somewhat stable.

Electrical conductivity is a measure of a substance’s ability to transmit electrical current, and the presence of foreign particles in water greatly affects conductivity. Figure 1b shows the conductivity variation of PAWs. Within 5 min, the conductivity of the PAW drastically increased. The primary reason for this significant increase was that a large number of active substances such as ROS and RNS (which are generated in the process of ionizing air dissolving in water) were formed [28].

### 2.2. NO_2_^−^ and NO_3_^−^ Concentration Changes of PAW

The detection of NO_2_^−^ and NO_3_^−^ in PAW is the primary indication for the formation of RNS. Their antibacterial activities have attracted considerable interest [29,30]. The changes of NO_2_^−^ and NO_3_^−^ concentrations in PAW are shown in Figure 2a,b. With the increase of plasma activation time, the concentration of NO_2_^−^ steadily increased, which was mainly due to the reaction of RNS substances such as NO, NO_2_, etc., generated by ionized air dissolved in water with OH^−^ and H_2_O. NO_2_^−^ can inhibit bacteria and protect color, and has a significant inhibitory effect on both Listeria monocytogenes and Clostridium botulinum [31]. Additionally, the concentration of NO_3_^−^ in PAW also rose gradually, showing similar changing rules as were observed with respect to NO_2_^−^. Shen et al. [32] showed that NO_3_^−^ was a secondary product in PAW, mainly from the reaction of NO_2_ and NO_2_^−^ with OH^−^ and H_2_O_2_ produced by the ionized gas. It has been proven that NO_3_^−^ expands the bacteriostatic activity of PAW to a certain extent and positively contributes to the sterilization effect in synergy with NO_2_^−^ [33].

### 2.3. H_2_O_2_ and O_2_^−^ Concentration Changes of PAW

H_2_O_2_ and O_2_^−^ are the main ROS in PAW and play an important role in antibacterial activity. The concentration changes of H_2_O_2_ and O_2_^−^ are shown in Figure 2c,d. The concentration of H_2_O_2_ increased significantly with the extension of treatment time, to 5372.69 μmol/L at 5 min, which was about 110 times higher than at 1 min. In contrast to the variable rule of H_2_O_2_ concentration, O_2_^−^ concentration showed a trend of rapid increase followed by a rapid decline. After 2 min treatment, O_2_^−^ concentration reached the highest value of 33,727.93 μmol/L and then dropped sharply to 4054.05 μmol/L after 5 min treatment. Although O_2_^−^ has considerable oxidative activity, it causes less intracellular damage, as it is thought to be unable to penetrate the cell membrane [34]. However, O_2_^−^ can be converted to hydrogen peroxide radicals (HOO∙) in PAW ($H^{+}+O_{2}^{-}↔\;\mathrm{HOO}∙$), which can damage intracellular components by penetrating the cell membrane.

As shown in Figure 1a, when the treatment time was 3 min, the pH of the PAW dropped to below 4.8 (the acid hydrolysis constant (pKa) of the reaction is 4.8) and a large amount of O_2_^−^ was converted into HOO∙ [35]. In addition, excessive H^+^ in the solution led to the conversion of a large amount of O_2_^−^ into H_2_O_2_ (2H++2O2−→O21+H2O2) [36]. The above two reactions could satisfactorily explain the reason for the sharp decline of O_2_^−^ after the treatment time reached 3 min.

### 2.4. Visual Assessment of Strawberry during Storage

As shown in Figure 3b, fresh strawberries without any treatment (control group) showed signs of surface spoilage on day 2 of storage at 20 °C After 4 days of storage, a substantial number of grey fungi appeared and the strawberries completely lost their edibility. Almost no visible spoilage was detected on the strawberries treated with PAW, particularly in the PAW 3 and PAW 2 treatment groups, and the strawberries still had good cosmetic qualities up to day 6, indicating that PAW inhibited the growth of fungi, in addition to bacteria. During the 6-day storage period, there were no discernible differences in visual preservation between PAW 2 and PAW 3.

### 2.5. Strawberry Quality Changes

Table 1 illustrates the effects of PAW treatment on the hardness and color of the strawberries. After 3 min treatment, the surface hardness, L*, a*, and b* of strawberries did not differ significantly from the fresh group (*p* > 0.05). In addition, when ∆E* > 3, it indicated that the color difference could be perceived; when ∆E* < 3, the color difference was not obvious [37]. In addition, pH value, an important parameter for sensory evaluation, was not significantly different between PAW 2 and PAW 3 groups, as compared to the fresh group (*p* > 0.05). On the other hand, the TSS, VC, and antioxidant capacity of the strawberries are essential markers for determining the nutritional quality of strawberries. As shown in Table 2, the TSS and VC content of the PAW-treated groups were not significantly different from the untreated groups (*p* > 0.05), indicating that PAW treatment did not affect TSS and VC content. Furthermore, the antioxidant capacity of treated strawberries was comprehensively evaluated by measuring DPPH radical scavenging activity and total antioxidant capacity. Table 2 shows that there was no significant difference among the treatment group (*p* > 0.05), indicating that PAW treatment did not significantly reduce the antioxidant capacity of strawberries. Therefore, it was concluded that PAW treatment had no significant adverse effects on strawberry quality.

The sensory and nutritional changes between 0 and 4 days of storage are illustrated in Table 1 and Table 2. The hardness of all samples decreased significantly with the increase of storage time (*p* < 0.05). The hardness was higher in the PAW treatment group compared to the control group, with the PAW 2 treatment group showing the slowest decreasing trend. As expected, the ∆E* value of the control group and PAW 0 reached 9.56 and 11.8. respectively, after 4 days of storage, indicating that the strawberries had lost their commercial value. The degree of color reduction after PAW treatment was much less than that of the control group and PAW 0. Apart from the PAW 1 treatment group, the ∆E* value of the PAW 2 and PAW 3 treatment groups were controlled at 6.21 and 8.08, respectively. In addition, the content of MDA in strawberries increased with the extension of storage time due to the decrease of antioxidant levels in strawberry tissue cells. Based on the above analysis, the overall quality decline rate of the PAW treatment group was lower than that of the untreated group during storage, indicating that PAW treatment had a positive effect on strawberry preservation.

### 2.6. Optimal Plasma Processing Condition Selection Using Multivariate Analysis Techniques

#### 2.6.1. Hierarchical Cluster Analysis

HCA was used to process the data matrix of each quality index of strawberry before and after treatment. HCA is a multivariate statistics technique that gradually categorizes data based on the similarity of the specified data matrix [38]. The result of the heat map with HCA operation is depicted in Figure 4. Using the color scale as a reference, various color blocks represent distinct values of relevant variables.

As can be seen from Figure 4, HCA divided the results of each processing group into two categories, with C-0, PAW 1-0, PAW 2-0, and PAW 3-0 sorted into one category, which visually indicated that the PAW treatment had no significant adverse effect on fresh strawberries. In contrast, the control and experimental groups, after 4 days of storage, were classified into another category, indicating that the quality of strawberries decreased to varying degrees after 4 days of storage. According to the tree diagram, the three groups of C-4 data and PAW 0-4 data were merged first, and then the three groups of PAW 3-4 data were later integrated into the same cluster. On the other hand, another three groups of PAW 2-4 data and PAW 1-4 data were ordered into another cluster. This indicated that, after 4 days of storage, the strawberry quality indexes of the control group and PAW 0 were similar, which was consistent with the analysis above. The heat map showed that the data from the PAW 3 group were closest to the quality indicators of the control group and PAW 0 samples. However, the current analysis could not directly analyze the difference between PAW 2 and PAW 1, and further analysis was required.

#### 2.6.2. Principal Component Analysis

To better explain and comprehend the quality variation of strawberries throughout storage, PCA analysis was performed on the storage start (0 days) and storage end (4 days) data in Table 1 and Table 2. PCA is a method of unsupervised statistical analysis that allows the raw data matrix to be compressed and downscaled to perform data analysis in a more parsimonious manner [39]. The feasibility of PCA analysis was first validated (KMO = 0.759; Chi-Square = 217.6; *p* < 0.05) using a correlation coefficient matrix to extract data for each experimental group for days 0 and 4 in Table 1 and Table 2 above, and the data were analyzed using the maximum variance method. Table 3 shows the total variance in PCA analysis. Two principal components with eigenvalues greater than 1 could be extracted, among which the eigenvalue of the first principal component (PC1) was 5.263 and that of the second principal component (PC2) was 1.427, which could interpret 52.631% and 14.272% of the original data, respectively. The remaining principal components (PC4-PC10) were all less than 1 and were excluded from the statistical analysis [40].

Loading variable plot (Figure 5a) showed that PC1 was positively correlated with hardness, L* value, DPPH radical scavenging activity, and total antioxidant capacity, and negatively correlated with MDA content, accounting for 52.63% of the total variance, while PC2 was positively correlated with TSS, explaining 14.27% of the total variance. PC1 and PC2 together explained 66.90% of the total variance. As shown in Figure 5b, the principal component score plots for the different experimental groups were compared after 0 and 4 days of storage. To get a clear view of the changes in the quality of strawberries in each treatment group during the storage period, Figure 5b was decomposed into five separate principal component score plots, corresponding to five treatments (Figure 5c–g). After 4 days of storage, PAW 2 (Figure 5f) showed minimal migration on PC1 and PC2 after storage (ΔPC1 = 0.78, ΔPC2 = 0.15), confirming the excellent preservation effect exerted by PAW2.

In the absence of significant effects of the above PAW treatments on strawberry quality, the migrations of PC1 and PC2 in each treatment group after storage were much lower than that of the experimental group with controls, indicating that PAW treatment had a maintenance effect on the freshness of strawberries. This also indicated that the PAW treatment had no significant adverse effects on the strawberry quality. PAW treatment had a certain delaying effect on the declining quality of strawberries, and the PAW 2 treatment in this study led to the best preservation effect. Under this treatment, the concentrations of the main oxidation active substances, including NO_2_^−^, NO_3_^−^, H_2_O_2_, and O_2_^−^, in the PAW 2 treatment group reached 0.397 mg/L, 35.612 mg/L, 1323.13 μmol/L, and 33,727.93 μmol/L, respectively. Among them, O_2_^−^ concentration was the highest in the treatment groups. O_2_^−^ can inhibit the activity of enzymes in the strawberry metabolism process and reduce the production of ethylene (which has a ripening effect in strawberries), allowing for optimal strawberry preservation.

## 3. Material and Methods

### 3.1. Materials and Chemicals

Fresh milk strawberries were purchased from Dandong, Liaoning province, and stored in a refrigerator at 4 °C in the laboratory. Trichloroacetic acid, thiobarbiturate acid, 1,1-diphenyl-2-trinitrophenylhydrazine, anhydrous ethanol, sodium dihydrogen phosphate, disodium hydrogen phosphate, p-aminobenzene sulfonamide, sodium nitrite, N-1-Naphthylethylenediamine Dihydrochloride, concentrated hydrochloric acid, sulfamic acid, and sodium nitrate were purchased from Aladdin Industries (Shanghai, China). The hydrogen peroxide detection kit and superoxide anion free radical detection kit were acquired from Shanghai Yuanye Biotechnology Co., Ltd. (Shanghai, China). The total antioxidant capacity detection kit was provided by Biyuntian Biotechnology Co., Ltd. (Shanghai, China), and the VC test strip was purchased from Merck (Darmstadt, Germany).

### 3.2. Plasma Device and Preparation of PAW

The atmospheric pressure DBD cold plasma system is shown in Figure 6a. The DBD plasma system configuration used in this study consisted of a DBD plasma reactor, a high-frequency AC power supply (CTP-2000 K, Nanjing Suman Electronics Co., Ltd., Nanjing, China), and an oscilloscope (TBS1107, Tektronix Inc., Beaverton, OR, USA, bandwidth 1 GHz). The plasma reactor was composed of two circular high-voltage and low-voltage steel electrodes with diameters of 50 mm and a circular upper quartz glass dielectric barrier with a diameter of 102 mm and a thickness of 1 mm. Figure 6b shows the chemical reaction generated by DBD plasma at the gas–liquid interface. In this study, the power supply was able to provide a sinusoidal wave output with an operating input voltage of 50 V and an input current of 1 A. Figure 6c shows the typical discharge voltage and current waveforms.

In this case, PAW was prepared by generating plasma on the water surface using atmospheric pressure air as the working supply gas at room temperature. The distance between the quartz plate and the water surface was 3 mm. Deionized water was activated by plasma for 0, 1, 2, 3, 4, and 5 min to prepare 10 mL of PAW, respectively. To make it easy to understand and define, PAW 1, PAW 2, PAW 3, PAW 4, and PAW 5 were denoted as referring to PAW after 1, 2, 3, 4, and 5 min of plasma activation, respectively. Additionally, PAW 0 was used to refer to water without plasma activation.

### 3.3. Pretreatment of Strawberries

Strawberries—with basically the same color and appearance, without external mechanical damage, complete, and undecomposed—were selected for the study by means of a colorimeter. The strawberries were placed in a cool and dry place after leaf removal and weighing. The strawberries were randomly divided into 5 groups, with 12 strawberries in each group. For the control group, strawberries were not subjected to any soaking treatment. For the other 4 treatment groups, the strawberries were immersed in PAW 0, PAW 1, PAW 2, and PAW 3 (duration time of 20 min), respectively. After pretreatment, the strawberries were moved to an incubator with a constant temperature and humidity (temperature: 20 ± 2 °C, relative humidity: 70 ± 5%) for 6 days. The strawberries were photographed and the quality indexes of strawberries were measured every 2 days.

### 3.4. Measurement of Physicochemical Properties of PAW

Both pH and conductivity were measured immediately after different plasma activation times for deionized water. The pH was measured at 25 ± 1 °C with a pH meter (Mettler-Toledo, Greisensee, Switzerland). The electrical conductivity was determined by a conductivity meter (DDS-307, Leici, Shanghai, China). The concentration of H_2_O_2_ in PAW was quantified by a hydrogen peroxide assay kit [41]. Similarly, the concentration of superoxide anion (O_2_^−^) in PAW was measured by a superoxide anion radical assay kit. The level of NO_3_^−^ and NO_2_^−^ in PAW were measured by the spectrophotometric method, as described by [32].

### 3.5. Quality Analysis of Strawberry

#### 3.5.1. pH Measurement

Using a knife, a strawberry sample that weighed 15 g was taken. The juice was extracted from the sample and ground in a self-sealing bag. After the pH meter was calibrated, the electrode was submerged in the sample solution, and the data was recorded after the reading had stabilized. Each measurement was repeated three times for each sample and the average value was taken as the pH value of the sample solution. The method was based on that of [14], with appropriate modifications. Samples of strawberries were taken and the hardness of the pulp at each location was determined by a texture analyzer (TA.XTplusC, Stable Micro System, London, UK) at three equally-spaced locations around the equatorial part of the strawberries.

#### 3.5.2. Firmness

The method was based on that of [14], with appropriate modifications. Samples of strawberries were taken and the hardness of the pulp at each location was determined by a texture analyzer (TA.XTplusC, Stable Micro System, London, UK) at three equally-spaced locations around the equatorial part of the strawberries.

#### 3.5.3. Color Quality

The color of the strawberries was characterized by a colorimeter (CR-400, Konica Minolta Inc., Tokyo, Japan) at room temperature, and the chroma value was expressed by the CIE system. For calibration, a standard whiteboard (L* = 95.35, a* = 0.01, b* = 2.30) was used before the measurements. The color value was expressed as L^*^ (+brightness, −darkness), a* (+red, −green), and b* (+yellow, −blue). Total color difference ∆E reflected the total color difference of the sample, calculated by the following formula [42]:(1)$$∆E=\sqrt{{\left(L^{*}-{\;L}_{0}^{*}\right)}^{2}+{\left(a^{*}-a_{0}^{*}\right)}^{2}+{\left(b^{*}-b_{0}^{*}\right)}^{2}}$$
where L_0_***, a_0_*, and b_0_* are chromaticity values of the untreated group, and L*, a*, and b* are chromaticity values of the treated group.

#### 3.5.4. Total Soluble Solids

The TSS levels of the strawberries were evaluated using a handheld Brix meter (Atago, Co., Ltd., Tokyo, Japan) at room temperature. A drop of juice obtained from a strawberry sample was placed on the handheld Brix meter to measure the soluble solids content. Before each measurement, a zero calibration was performed with deionized water, and the measurement was repeated three times for each sample.

### 3.6. Malondialdehyde

Strawberry samples (1 g) were homogenized with 5 mL of trichloroacetic acid (100 g/L). The homogenate was centrifuged at 10,000 rpm for 10 min at 4 °C. Subsequently, 2 mL of the supernatant was taken, and 2 mL of thiobarbituric acid (0.67%, *w*/*v*) was added to the supernatant. The mixed solution was heated in distilled water at 100 °C for 20 min, then removed and cooled. The supernatant was taken and its absorbance was recorded at 450, 532, and 600 nm [14]. The MDA content was calculated based on the following equation:(2)$$C\;\left(\mathrm{mmol}/g\right)=6.45\;\times \;\left({\mathrm{OD}}_{532}-{\mathrm{OD}}_{600}\right)-\;0.56\;\times \;{\mathrm{OD}}_{450}$$

### 3.7. Vitamin C Content

1 mL of freshly squeezed strawberry juice was diluted to 5 mL and the VC content of the strawberry juice was measured using an RQflex20 reflectometer. The VC test strip was immersed in the diluted sample solution for 5 s, then taken out and shaken off the surface. The test strip was inserted into the reflectometer and the VC content (mg/L) was read directly. The VC content was calculated based on the following equation:(3)$$V_{C}\left(\mathrm{mg}/L\right)={\;C}_{0}\;\times \;N$$
where C_0_ is the VC content (mg/L) of the test strip and N is the sample dilution.

### 3.8. Determination of Antioxidant Activities

#### 3.8.1. Total Antioxidant Capacity

The total antioxidant capacity of strawberries was assessed using the total antioxidant capacity (ABTS) kit. ABTS working solution (200 μL) was added to each 96-well enzyme labeled plate. Then, 10 μL of deionized water was added to the control well, and 10 μL of sample solution was added to the sample well. Next, 10 μL of Trolox series standard solution (0.15, 0.3, 0.6, 0.9, 1.2, and 1.5 mM) was added to the standard well, followed by incubation for 2–6 min at room temperature and measurement of A414 using an enzyme marker (Synergy2, BioTek, model S, Winooski, VT, USA). The standard curve was plotted using the Trolox series standard solution as the horizontal coordinate and the absorbance value as the vertical coordinate to obtain the regression equation. The absorption value of the sample was substituted into the regression equation to obtain the total antioxidant capacity of the sample, which was calculated by the following formula:(4)$$\mathrm{Antioxidant}\;\mathrm{capacity}\;\left(\mathrm{mmol}/L\right)={\;C}_{0}\;\times \;\mathrm{NVitaminC}\left(\mathrm{mg}/L\right)=C_{0}\times N$$
where C_0_ is the total antioxidant capacity (mmol/L) derived from the absorbance in the sample to be measured on the standard curve and N is the sample dilution.

#### 3.8.2. DPPH Radical Scavenging Activity

The DPPH radical scavenging activity of strawberries was determined according to the method in [43], with appropriate modification. Briefly, 1 mL of freshly squeezed strawberry juice was taken in a sterile tube and an appropriate amount of deionized water (4 mL) was added as the sample solution. Standard tubes, assay tubes, and control tubes were set, and then 1 mL of ethanol and 3 mL of DPPH solution were added to the standard tube. In the assay tube, 1 mL of the sample solution and 3 mL of the DPPH solution were combined. Then, 1 mL of the sample solution and 3 mL of ethanol were incorporated into the control tube. The three sets of tubes were protected from light for 30 min, after which the absorbance values of the standard, control, and assay tubes were determined using a UV spectrophotometer at room temperature (25 °C) at the UV absorption peak of 517 nm. These were recorded as A_0_, A_1_, and A_2_, correspondingly. The DPPH radical scavenging activity was calculated using the following equation:(5)$$\mathrm{DPPH}\;\mathrm{radical}\;\mathrm{scavenging}\;\mathrm{activity}\;(\%)=1-\frac{A_{1}-A_{2}}{A_{0}}\;\times \;100\%\;\mathrm{Antioxidantcapacity}\left(\mathrm{mmol}/L\right)=C_{0}\times \mathrm{NVitaminC}\left(\mathrm{mg}/L\right)=C_{0}\times N$$

### 3.9. Statistical Analysis

OriginLab 8.0 software (OriginLab Inc., Northampton, MA, USA) and SPSS software (version 20.0, IBM, Chicago, IL, USA) was adopted to perform statistical analysis. Statistical differences were determined using one-way or multifactorial analysis of variance (ANOVA) followed by the post hoc Turkey test. Additionally, the differences were further determined by the least significant difference test at *p* < 0.05 level. Each experiment was repeated in triplicate and the results were presented as mean values ± standard deviation. Multivariate data analyses, including principal component analysis (PCA) and hierarchical cluster analysis (HCA), were performed using SPSS.

## 4. Conclusions

The PAW prepared using DBD cold plasma treatment for 2 min was the most suitable for the preservation of strawberries. The physicochemical properties of PAW were related to PAW treatment time. The PAW treatment did not significantly damage the quality of the strawberries themselves. On the contrary, PAW treatment was able to delay the quality deterioration of strawberries during the storage period. Multivariate data analyses, including PCA and HCA, showed that the PAW 2 treatment had the best quality indicators after 4 days of storage, among which hardness, L* value, and antioxidant activities were the most significant. It was therefore concluded that PAW proved, in this work, to be a promising nonthermal technology for use in the maintenance of the quality of strawberries, as well as the extension of their shelf life. In the future, a more comprehensive understanding of the generation and reaction of short-lived ROS will be needed to elucidate the oxidation mechanism of the PAW method.

## Figures and Tables

**Figure 1 molecules-28-02677-f001:**
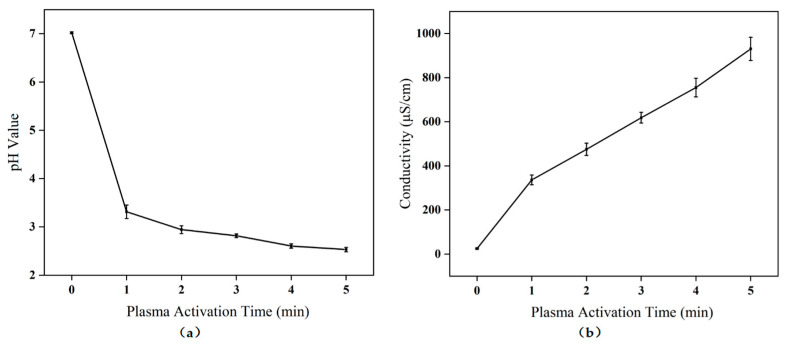
The values of (**a**) pH and (**b**) electrical conductivity with the plasma activation time increase.

**Figure 2 molecules-28-02677-f002:**
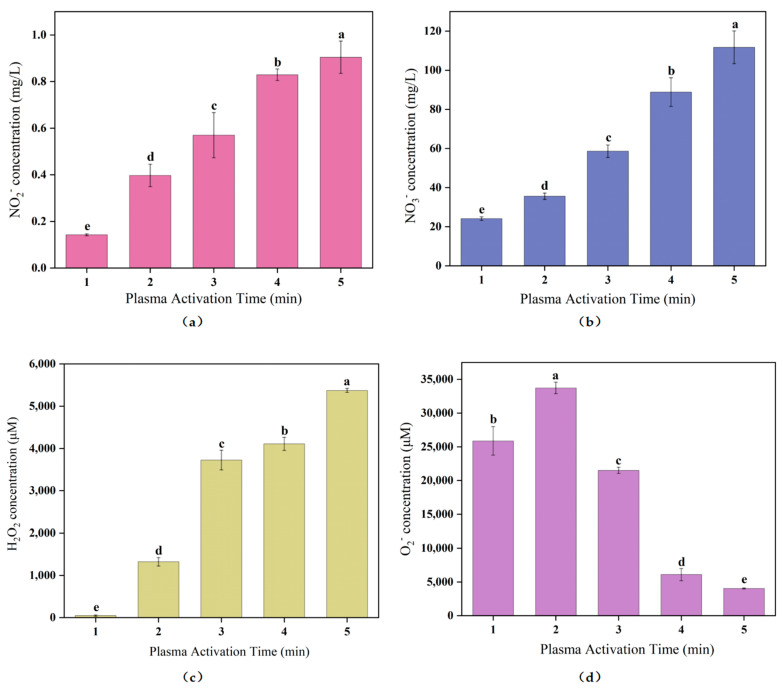
(**a**) NO_2_^−^ concentration; (**b**) NO_3_^−^ concentration; (**c**) H_2_O_2_ concentration and (**d**) O_2_^−^ concentration in PAW with the plasma activation time increase. Different letters (a–e) mean significant differences here.

**Figure 3 molecules-28-02677-f003:**
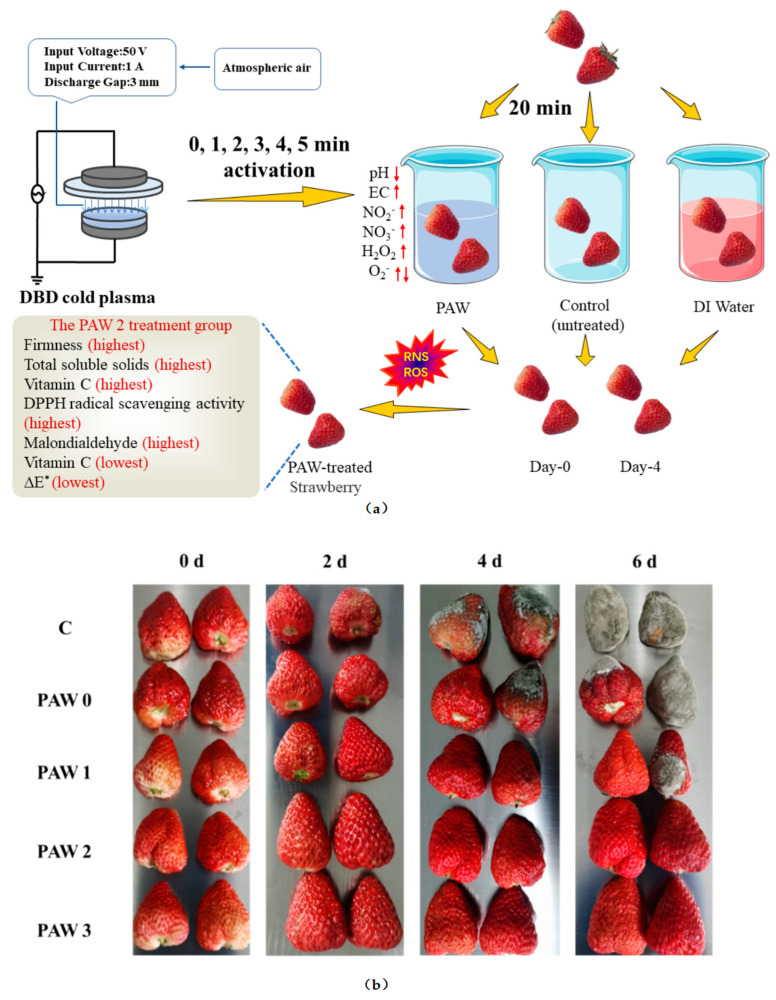
(**a**) Schematic diagram of the experimental arrangement for strawberry quality change assessment. (**b**) Effects of different PAW treatments on strawberry appearance.

**Figure 4 molecules-28-02677-f004:**
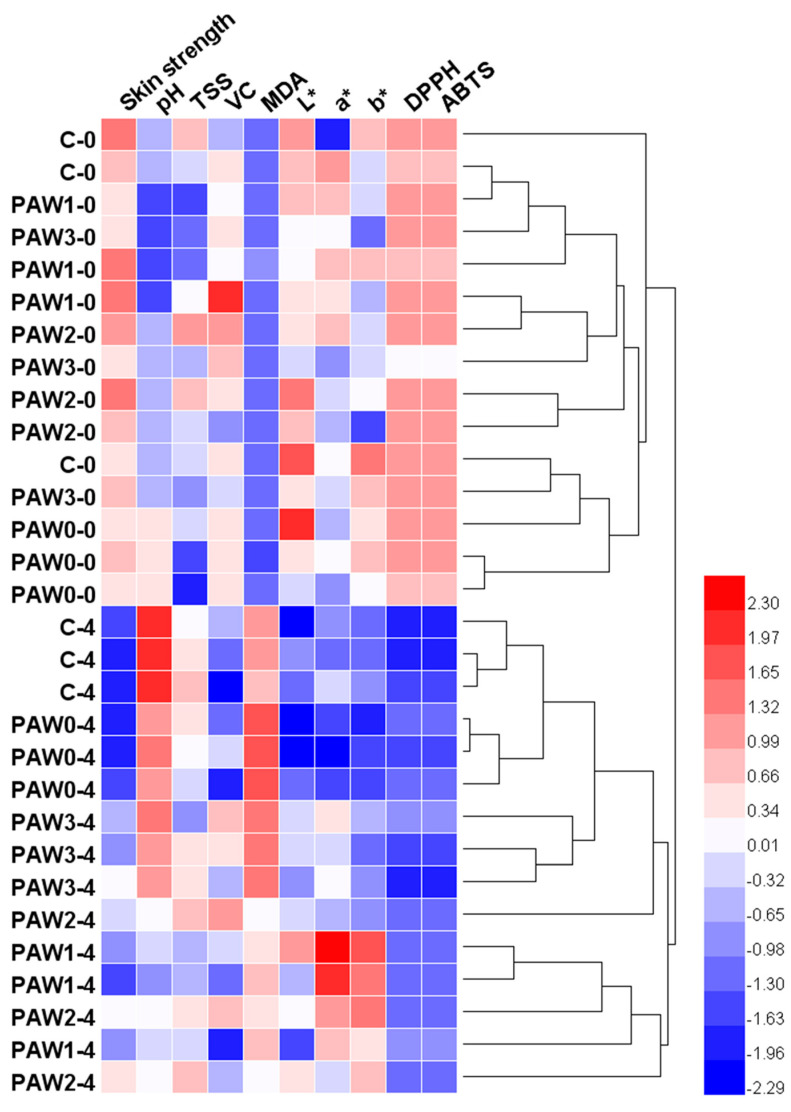
Heatmap combined with the dendrogram of cluster analysis based on the quality of strawberries.

**Figure 5 molecules-28-02677-f005:**
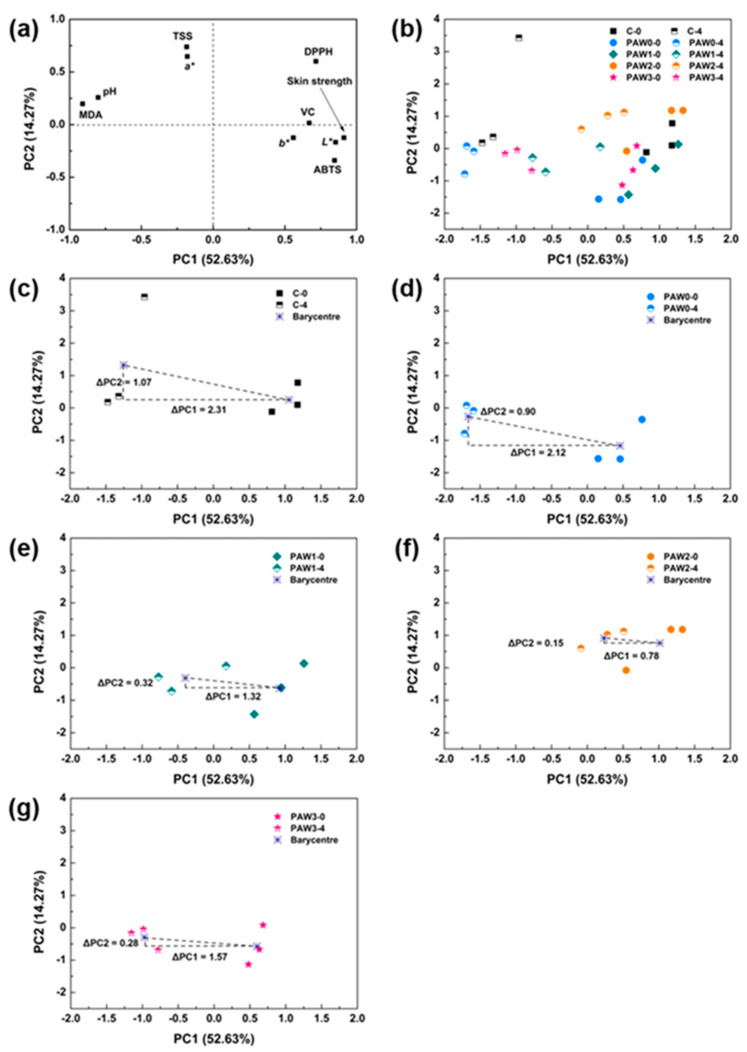
Principal component analysis (PCA) of the strawberry quality, comparing fresh strawberries, strawberries after PAW treatment (0 d), and strawberries after storage (4 d). (**a**) Loading variable plot of PCA. (**b**) Comprehensive PCA score plot comparing fresh strawberries to strawberries after 0 and 4 days when subjected to various treatments. Individual PCA score plots of no-treatment control (**c**), PAW 0 (**d**), PAW 1 (**e**), PAW 2 (**f**), and PAW 3 (**g**) for evaluation of the general quality changes of strawberries during storage, comparing quality after 0 and 4 days.

**Figure 6 molecules-28-02677-f006:**
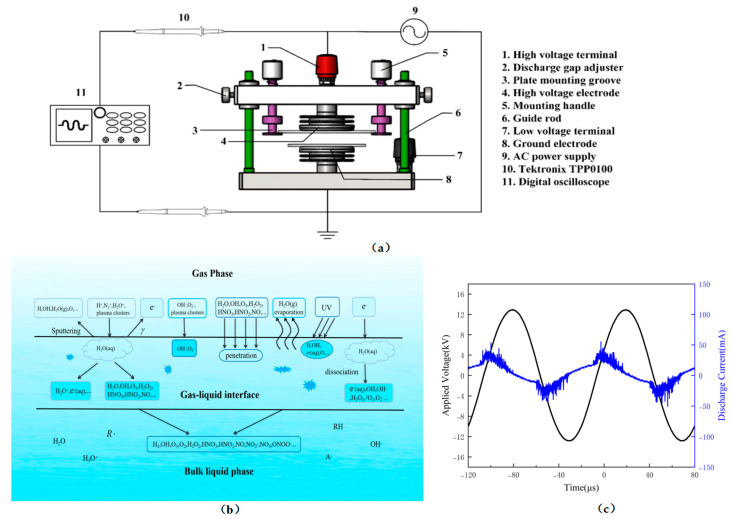
(**a**) Schematic diagram of experimental set-up for PAW generation; (**b**) chemical processes induced by DBD atmospheric cold plasma in gas–liquid interface; (**c**) typical discharge voltage and current waveforms.

**Table 1 molecules-28-02677-t001:** Sensory quality changes of the strawberry in different treatment groups during storage.

Treatments	Storage Time (Day)	Firmness (N)	pH	Color Parameters			
				L*	a*	b*	ΔE*
C	0	74.8 ± 7.10 ^a^	3.64 ± 0 ^b^	50.5 ± 1.97 ^a^	26.0 ± 2.14 ^a^	17.3 ± 2.22 ^a^	-
	4	28.8 ± 1.33 ^c^	3.82 ± 0.01 ^a^	43.3 ± 2.04 ^ab^	26.1 ± 0.91 ^b^	13.4 ± 0.69 ^b^	9.56 ± 1.46 ^ab^
PAW 0	0	68.9 ± 5.55 ^a^	3.71 ± 0 ^a^	50.0 ± 2.52 ^ab^	27.9 ± 2.25 ^a^	17.4 ± 2.53 ^a^	5.82 ± 1.92 ^a^
	4	27.3 ± 2.40 ^c^	3.76 ± 0.01 ^b^	41.3 ± 1.58 ^b^	23.7 ± 0.71 ^c^	11.5 ± 0.77 ^b^	11.8 ± 1.65 ^a^
PAW 1	0	77.9 ± 10.95 ^a^	3.58 ± 0 ^c^	46.8 ± 2.54 ^c^	27.2 ± 1.64 ^a^	15.7 ± 2.94 ^a^	6.89 ± 2.33 ^a^
	4	37.5 ± 6.81 ^b^	3.65 ± 0.03 ^d^	46.0 ± 3.49 ^a^	31.0 ± 1.91 ^a^	20.3 ± 2.20 ^a^	9.62 ± 0.3 ^ab^
PAW 2	0	77.3 ± 3.35 ^a^	3.65 ± 0 ^b^	48.2 ± 1.74 ^abc^	26.9 ± 1.29 ^a^	15.3 ± 1.69 ^a^	5.89 ± 1.18 ^a^
	4	60.4 ± 5.16 ^b^	3.68 ± 0.01 ^c^	47.2 ± 0.81 ^a^	27.5 ± 1.72 ^b^	17.7 ± 3.88 ^b^	6.21 ± 1.87 ^c^
PAW 3	0	68.8 ± 4.16 ^a^	3.63 ± 0.03 ^b^	47.3 ± 0.9 ^bc^	26.9 ± 1.23 ^a^	16.0 ± 2.78 ^a^	6.00 ± 1.82 ^a^
	4	49.8 ± 9.01 ^b^	3.76 ± 0.01 ^b^	45.9 ± 1.09 ^a^	27.5 ± 0.82 ^b^	13.8 ± 0.78 ^a^	8.08 ± 0.65 ^bc^

Note: Values with different letters within the same column are significantly different in each group (*p* < 0.05).

**Table 2 molecules-28-02677-t002:** Nutritional quality changes of strawberries in different treatment groups during storage.

Treatments	Storage Time (Day)	TSS (°Brix)	V_C_ (mg/L)	DPPH Radical Scavenging Activity (%)	Antioxidant Activity (mmol/L)	MDA Content (mmol/g)
C	0	10.2 ± 0.55 ^ab^	113.3 ± 12.5 ^a^	97.1 ± 1.04 ^a^	1.31 ± 0.02 ^a^	6.45 ± 0.09 ^a^
	4	10.5 ± 0.20 ^a^	113.3 ± 35.9 ^ab^	95.0 ± 2.53 ^ab^	0.61 ± 0.08 ^b^	8.58 ± 0.21 ^c^
PAW 0	0	9.00 ± 0.75 ^b^	120.3 ± 8.0 ^a^	93.6 ± 1.22 ^b^	1.31 ± 0.02 ^a^	6.41 ± 0.18 ^a^
	4	10.2 ± 0.45 ^a^	89.6 ± 16.17 ^a^	92.2 ± 0.62 ^b^	0.71 ± 0.03 ^ab^	9.53 ± 0.10 ^a^
PAW 1	0	9.48 ± 0.97 ^ab^	127.3 ± 24.0 ^a^	95.8 ± 1.49 ^a^	1.31 ± 0.03 ^a^	6.53 ± 0.24 ^a^
	4	9.63 ± 0.15 ^b^	90.8 ± 16.7 ^ab^	94.9 ± 1.78 ^ab^	0.77 ± 0.03 ^a^	8.32 ± 0.13 ^c^
PAW 2	0	10.5 ± 0.56 ^a^	107.3 ± 24.1 ^a^	98.0 ± 1.48 ^a^	1.31 ± 0.01 ^a^	6.47 ± 0.14 ^a^
	4	10.6 ± 0.21 ^a^	119.0 ± 18.2 ^b^	97.1 ± 2.00 ^a^	0.75 ± 0.03 ^ab^	7.93 ± 0.11 ^d^
PAW 3	0	9.10 ± 0.34 ^b^	117.0 ± 10.1 ^a^	95.6 ± 2.13 ^ab^	1.26 ± 0.12 ^a^	6.41 ± 0.05 ^a^
	4	10.5 ± 0.12 ^a^	113.0 ± 13.5 ^ab^	91.8 ± 0.71 ^b^	0.66 ± 0.11 ^ab^	9.07 ± 0.07 ^b^

Note: values with different letters within the same column are significantly different in each group (*p* < 0.05).

**Table 3 molecules-28-02677-t003:** Total variance explained in PCA.

Component	Initial Eigenvalues		
	Eigenvalue	% of Variance	Cumulative (%)
1	5.263	52.631	52.631
2	1.427	14.272	66.903
3	0.996	9.963	76.866
4	0.895	8.951	85.817
5	0.571	5.709	91.526
6	0.407	4.065	95.591
7	0.158	1.581	97.172
8	0.138	1.377	98.549
9	0.098	0.980	99.528
10	0.047	0.472	100.000

## Data Availability

Not applicable.

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
