# Peer review of "Physicochemical Properties of Plasma-Activated Water and Its Control Effects on the Quality of Strawberries"

_molecules, 2023, doi:10.3390/molecules28062677_

Round 1

Reviewer 1 Report

The manuscript entitled “Physicochemical properties of plasma-activated water and its control effects on the quality of strawberry” aims to investigate the effects of plasma-activated water (PAW) generated by dielectric barrier discharge cold plasma at the gas-liquid interface on the quality of fresh strawberries during storage. The manuscript is very interesting, and the results have a high potential for application. The manuscript is very well prepared, and reading it was a real pleasure. My specific comments are given below.

The abstract is suitable.

The introduction is informative enough and provides all the necessary elements.

Material and methods are given in detail.

Line 113: “Strawberries with basically the same color” – the authors should specify how they determined this.

Sections 2.5.4. and 2.6. subtitles should be given in full, no matter that the abbreviation was already introduced. It would be more evident.

Results and discussion are well presented in accordance with the scientific principles. The discussion is scientifically sound and with respect to the existing literature.

The conclusion is well-written and in accordance with the presented results.

I suggest that the manuscript be accepted after addressing the minor issues mentioned above.  

Reviewer 2 Report

The manuscript submitted for review raises an important issue related to the possibilities of extending the freshness of strawberries, which are among the least durable fruits.

I have some comments that are as follows:

line 31 - "internal and external" - what? Seems like sth was missing

line 61 - references should be added after the sentence ending "producta treated with PAW"

line 96 - something is wrong with numbers of figures, whould it be Figure 1b?

line 99 - Figure 1 c?

line 109- should it be Figure 1 instead of Figure 4?

lines 118-119 - the authors state that strawberries were immersed in PAW 0, PAW 1, PAW 2 and PAW 3. What about PAW 4 and PAW 5 mentioned in line 105?

subheadings 2.5.4., 2.6, 2.7, 3.6.1 and 3.6.2 - I would suggest to use full names, not abbreviations in the headings

line 222 - agains som problems with numbers of Figures, should it be Figure 2a instead of 5a?

line 230 - figure 2 b instead 6b?

line 235 - Figure 2?

Tables 1 and 2 - according to the methods described the measurements were done on days 0, 2, 4 and 6, why only results for days 0 and 4 are presented?

The page with tables is wrongly formatted, and the following pages have wrong page numbers.

section 3.6.2. - why days 0 and 4 only?
